# GS-VTON: Controllable 3D Virtual Try-on with Gaussian Splatting

## Abstract

Diffusion-based 2D virtual try-on (VTON) techniques have recently demonstrated strong performance, while the development of 3D VTON has largely lagged behind. Despite recent advances in text-guided 3D scene editing, integrating 2D VTON into these pipelines to achieve vivid 3D VTON remains challenging. The reasons are twofold. First, text prompts cannot provide sufficient details in describing clothing. Second, 2D VTON results generated from different viewpoints of the same 3D scene lack coherence and spatial relationships, hence frequently leading to appearance inconsistencies and geometric distortions. To resolve these problems, we introduce an image-prompted 3D VTON method (dubbed GS-VTON) which, by leveraging 3D Gaussian Splatting (3DGS) as the 3D representation, enables the transfer of pre-trained knowledge from 2D VTON models to 3D while improving cross-view consistency. **(1)** Specifically, we propose a personalized diffusion model that utilizes low-rank adaptation (LoRA) fine-tuning to incorporate personalized information into pre-trained 2D VTON models. To achieve effective LoRA training, we introduce a reference-driven image editing approach that enables the simultaneous editing of multi-view images while ensuring consistency. **(2)** Furthermore, we propose a persona-aware 3DGS editing framework to facilitate effective editing while maintaining consistent cross-view appearance and high-quality 3D geometry. **(3)** Additionally, we have established a new 3D VTON benchmark, *3D-VTONBench*, which facilitates comprehensive qualitative and quantitative 3D VTON evaluations. Through extensive experiments and comparative analyses with existing methods, the proposed GS-VTON has demonstrated superior fidelity and advanced editing capabilities, affirming its effectiveness for 3D VTON.

## 1 Introduction

Driven by advancements in neural rendering, virtual try-on (VTON) techniques represent a significant milestone in the intersection of fashion and computer vision. These technologies are increasingly utilized across various domains, such as online shopping (Kim & Forsythe, 2008; Zhang et al., 2019), VR/AR avatar modeling (Mystakidis, 2022), and gaming (Lerner et al., 2007), enabling users to visualize how different garments will look on them without the need for a physical try-on. Traditional methods (Han et al., 2018; Wang et al., 2018; Meng et al., 2010; Hauswiesner et al., 2013; Hsieh et al., 2019) for this task primarily emphasize 2D image editing. Typically, they achieve virtual try-on by estimating pixel displacements using optical flow (Canny, 1986) and employing pixel warping techniques to seamlessly blend clothing with the individual. However, these 2D VTON approaches have struggled with occlusion issues and have difficulty accommodating complex human poses and clothing. With the rise of deep learning, methods (Choi et al., 2021; Ge et al., 2021a;b; Lee et al., 2022; Men et al., 2020) utilizing Generative Adversarial Networks (GANs) (Goodfellow et al., 2014) have been introduced, aiming for more effective virtual fitting experiences. Despite their promise, these methods face challenges when handling custom user images that fall outside the training data. Although approaches (Zhu et al., 2023; Choi et al., 2024; Kim et al., 2024; Xu et al., 2024) leveraging large language models (Radford et al., 2021b) and diffusion models (Song et al., 2021; Stability.AI, 2022) have demonstrated improved performance and generalization, these approaches still struggle with generating consistent multi-view images and accurately modeling 3D representations of garments.

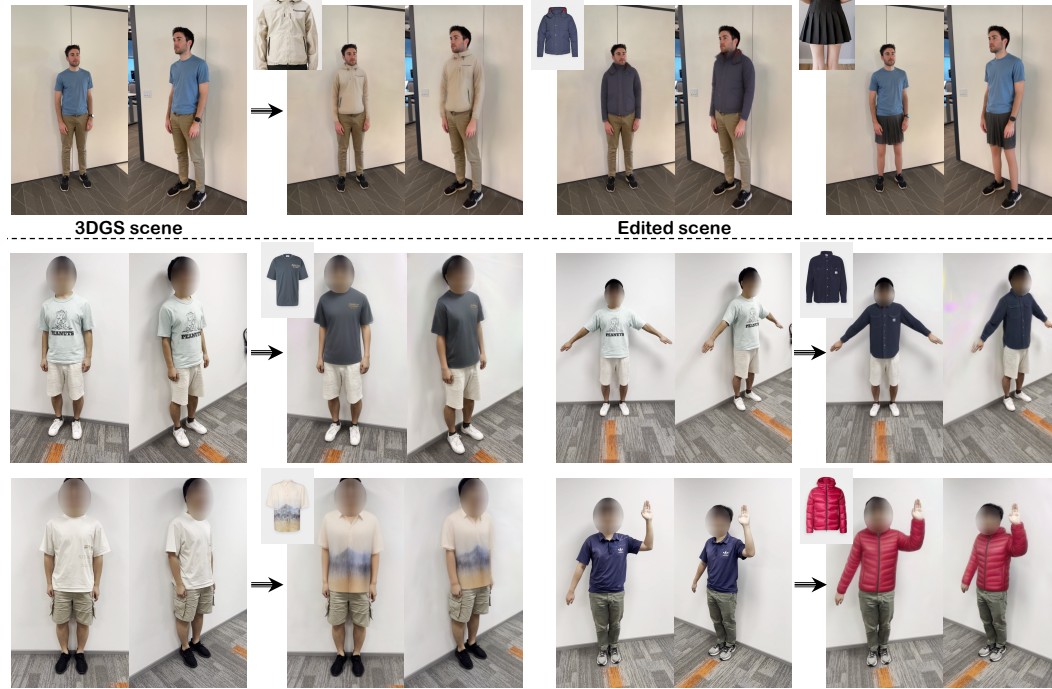

Figure 1: **Examples of 3D virtual try-on results obtained via GS-VTON.** Our approach facilitates high-fidelity editing of 3D garments, featuring intricate geometry and texture, under various scenarios with diverse cloth types, body shapes, and poses.

Recently, neural radiance field (NeRF) (Mildenhall et al., 2021) and 3D Gaussian Splatting (3DGS) (Kerbl et al., 2023) have garnered significant attention for their efficient differentiable rendering capabilities, sparking research into text-guided 3D editing algorithms (Haque et al., 2023; Cyrus & Ayyan, 2023; Wu et al., 2024). Instruct-NeRF2NeRF (Haque et al., 2023) leverages a pre-trained diffusion model to edit rendered images while computing image-level loss based on textual prompts, allowing gradients to be back-propagated for modifying 3D differentiable scenes. Following this, subsequent research efforts (Zhuang et al., 2023; Shao et al., 2023; Dong & Wang, 2024; Cheng et al., 2023; Han et al., 2023; Zhou et al., 2024b) have aimed to improve quality and broaden the applications of Instruct-NeRF2NeRF across various tasks. However, these methods generally apply global edits to the 3D scene, limiting their effectiveness for VTON applications. While GaussianEditor (Chen et al., 2023b) and TIP-Editor (Zhuang et al., 2024) have been developed to facilitate local editing, they still encounter difficulties when modifying clothing items based solely on textual descriptions (see Fig. 5). In addition, the rising use of image prompts in VTON applications, which convey richer information than text, underscores the urgent need for adaptable 3D VTON methods that accommodate user-specified images. On the other hand, directly applying 3D editing algorithms with diffusion-based 2D VTON models often leads to unsatisfactory results, primarily due to two major limitations. *First,* current 2D VTON diffusion models struggle to accurately visualize how the input clothing image would appear from different viewpoints, resulting in multi-view inconsistencies within the edited 3D scene. This issue stems from a lack of coherence and spatial relationships. Furthermore, since we aim to modify individual garments rather than the entire body, maintaining consistency with other body parts becomes even more challenging. *Second,* existing 2D VTON diffusion model may still yield suboptimal results when dealing with data that falls outside their training distribution, leading to issues such as blurriness and distortions in both appearance and geometry.

To address this challenge, we present a novel image-prompted 3D VTON method in this paper, entitled **GS-VTON**, which could achieve fine-grained editing of human garments. By taking a garment image and multi-view human images as input, our method comprises two major components, personalized diffusion model via LoRA fine-tuning and persona-aware 3DGS editing, to achieve this objective. *First,* we enhance the pre-trained 2D VTON diffusion model by incorporating personalized

information through a low-rank adaptation (LoRA) module. This enhancement allows the model to better reflect the specific characteristics of the input data by extending its learned distribution. *Second,* we introduce a reference-driven image editing approach that can simultaneously edit multi-view images while maintaining high consistency. This method forms a robust foundation for effectively training the LoRA module. *Third,* we design a persona-aware 3DGS editing process that refines the original editing by blending two predicted attention features: one for editing and the other for ensuring coherence across different viewpoints. This strategy facilitates effective editing while enhancing multi-view consistency in geometry and texture.

Moreover, to support more thorough qualitative and quantitative evaluations, we establish a 3D VTON benchmark, named *3D-VTONBench*, which, to our knowledge, is the first dataset of its kind. As presented in Fig. 1, our method achieves high-fidelity 3D VTONs across diverse scenarios with various garments and human poses. Comprehensive comparisons with existing techniques also demonstrate that our approach significantly surpasses existing methods, establishing a new state-of-the-art in 3D VTON.

Our contributions could be summarized as follows:

- We introduce GS-VTON that, by extending the 2D pre-trained virtual try-on diffusion model to 3D, can take garment images as input to perform fine-grained 3D virtual try-on.

- To enhance multi-view consistency, we propose a reference-aware image editing technique that simultaneously generate consistent multi-view edited images, as well as a persona-aware 3DGS editing which takes into account both the intended editing direction and the original set of edited images.

- We have created the first benchmark for 3D virtual try-on, enabling more comprehensive evaluations. Extensive experiments demonstrate that our method establishes a new state-of-the-art performance for 3D virtual try-on.

## 2 RELATED WORKS

**2D Diffusion-based Generative Model.** In recent years, there have been significant advancements in vision-language technologies, including methods like Contrastive Language-Image Pretraining (CLIP) (Radford et al., 2021a) and various diffusion models (Ho et al., 2020; Dhariwal & Nichol, 2020; Rombach et al., 2022b; Song et al., 2021). These models, trained on billions of text-image pairs, exhibit a strong understanding of real-world image distributions, enabling them to generate high-quality and diverse visuals. Such developments have greatly advanced the field of text-to-2D content generation (Saharia et al., 2022; Ramesh et al., 2022; Balaji et al., 2022; Stability.AI, 2022; 2023a) and text-to-video generation (Blattmann et al., 2023a; Liu et al., 2024; Guo et al., 2023; Ma et al., 2024; Huang et al., 2024). Following these techniques, subsequent research has focused on enhancing control over generated outputs (Zhang & Agrawala, 2023; Zhao et al., 2023; Mou et al., 2023), adapting diffusion models for video sequences (Singer et al., 2023; Blattmann et al., 2023c), facilitating both image and video editing (Hertz et al., 2022; Kawar et al., 2022; Wu et al., 2022; Brooks et al., 2023; Valevski et al., 2022; Esser et al., 2023; Hertz et al., 2023). Additionally, efforts have also been made to boost performance in personalized content generation (Ruiz et al., 2023a; Gal et al., 2023). Despite these advancements, the skill of crafting effective prompts remains crucial. Furthermore, in virtual try-on applications, which is the main target of this paper, textual descriptions frequently struggle to convey the intricate details of clothing as effectively as images, complicating the process of achieving realistic 2D virtual try-on.

**Image-based Virtual Try-on.** Image-based virtual try-on aims to create a visualization of a target person wearing a specific garment. Traditionally, methods (Choi et al., 2021; Lee et al., 2022; Men et al., 2020; Ge et al., 2021b; Xie et al., 2023; Ge et al., 2021a) based on generative adversarial network (GAN) (Goodfellow et al., 2014) have been proposed to correspondingly deform the garment before fitting it to the human subject. Subsequent efforts (Issenhuth et al., 2020; Lee et al., 2022; Ge et al., 2021b; Choi et al., 2021) have been made to minimize the discrepancies between the altered garment and the person. However, these methods are often constrained by the training dataset, showing limited generalization to images outside the pre-trained distribution. More recently, benefiting from the success of diffusion models (Saharia et al., 2022; Ramesh et al., 2022; Balaji et al.,

2022; Stability.AI, 2022), researches have explored applying them to tackle the existing limitations for virtual try-on. Specifically, TryOnDiffusion (Zhu et al., 2023) introduces a dual UNet architecture, demonstrating the potential of diffusion-based approaches when trained on extensive datasets; Yang et al. (2023) treats the virtual try-on as the exemplar-based image inpainting; Stableviton (Kim et al., 2024), Ladi-VTON (Morelli et al., 2023) and Gou et al. (2023) fine-tune diffusion models to achieve high-quality results; IDM-VTON (Choi et al., 2024) explores the usage of high-level semantics and low-level features to handle the task of identity preservation during virtual try-on. Despite showing promise, they can still yield suboptimal results for out-of-distribution data, and transferring pre-trained 2D knowledge directly to the 3D space remains challenging.

**3D Scene Editing.** Leveraging the advancement of differentiable 3D representation, *i.e.*, NeRF (Mildenhall et al., 2020) and 3DGS (Kerbl et al., 2023), and diffusion-based text-to-2D generation methods (Stability.AI, 2022; Brooks et al., 2023), text-driven 3D scene editing methods have emerged for modifying 3D subjects using diffusion models. Among them, Instruct-NeRF2NeRF (IN2N) (Haque et al., 2023) is the first to propose editing 2D renderings with Instruct-Pix2Pix (Brooks et al., 2023) and back-propagating gradients to adjust the 3D scene until convergence. While IN2N shows promise, it faces challenges such as instability, inefficient training, blurry results, and significant artifacts. These issues arise from the diffusion models' lack of 3D awareness, particularly regarding camera pose, leading to inconsistent multi-view rendering edits. To address these limitations, subsequent works (Po et al., 2024; Wang et al., 2024) have aimed to enhance performance from various angles: Instruct-Gaussian2Gaussian (Cyrus & Ayyan, 2023) replaces the 3D representation of NeRF with 3DGS and introduces improved dataset updating strategies for better training efficiency. Vica-NeRF (Dong & Wang, 2024) first selects several reference images from the input dataset, edits them using Instruct-Pix2Pix, and then blends the results for the remaining dataset to reduce inconsistencies. However, this blending does not fully resolve the consistency issue and often results in blurry edits for human subjects. DreamEditor (Zhuang et al., 2023) applies personalized DreamBooth (Ruiz et al., 2023b) to achieve local editing. TIP-Editor (Zhuang et al., 2024) introduces a 3D bounding box as a condition to enhance control over local editing. Despite promising results in adding objects to 3D scenes, these methods struggle with local modifications of internal geometry and textures. GaussianEditor (Chen et al., 2023b) utilizes large language models (Kirillov et al., 2023) for text-driven local editing. GaussCTRL achieves similar outcomes using a depth-conditioned ControlNet (Zhang & Agrawala, 2023). Unfortunately, existing techniques typically do not accept images as input and have difficulty performing garment editing for effective 3D virtual try-on. While GaussianVTON (Chen et al., 2024a) presents a three-stage editing pipeline aimed at a similar task, it may still face challenges in largely altering the original garment geometry.

## 3 METHODOLOGY

We present GS-VTON, a novel 3D virtual try-on method that enables controllable local editing to the human garment within a 3D Gaussian Splatting (3DGS) scene. Specifically, our method leverages multi-view human images $\mathcal{I}_{\text{train}}$, and a garment image as inputs to achieve this objective. In the subsequent sections, we first describe the preliminary knowledge that underpins our method in Sec. 3.1. We will then delve into the core elements of GS-VTON, which include (1) personalized inpainting diffusion model adaptation via reference-driven image editing and LoRA fine-tuning in Sec. 3.2, and (2) persona-aware self-attention mechanism for achieving customizable 3D virtual try-ons using 3DGS in Sec. 3.3. An overview of GS-VTON is illustrated in Fig. 2.

### 3.1 PRELIMINARIES

**3D Gaussian Splatting.** Unlike NeRF (Mildenhall et al., 2021), which employs neural networks to synthesize novel views, 3D Gaussian Splatting (3DGS) (Kerbl et al., 2023) takes another direction by directly optimizing the 3D position $\mathbf{x}$ and attributes of 3D Gaussians, i.e, opacity $\alpha$, anisotropic covariance, and spherical harmonic (SH) coefficients $\mathcal{SH}$ (Ramamoorthi & Hanrahan, 2001). Specifically, the 3D Gaussian $G(\mathbf{x})$ is defined by a 3D covariance matrix $\Sigma$ centered at point (mean) $\mu$:

$$G(\mathbf{x}) = e^{-\frac{1}{2}(\mathbf{x}-\mu)^T \Sigma^{-1} (\mathbf{x}-\mu)}. \tag{1}$$

Drawing inspiration from (Lassner & Zollhofer, 2021), 3DGS implements a tile-based rasterizer: The screen is first divided into tiles, such as $16 \times 16$ pixels. Each Gaussian is instantiated based on the

Figure 2: **Overview of GS-VTON.** We enable 3D virtual try-on by leveraging knowledge from pre-trained 2D diffusion models and extending it into 3D space. **(1)** We introduce a reference-driven image editing method that facilitates consistent multi-view edits. **(2)** We utilize low-rank adaptation (LoRA) to develop a personalized inpainting diffusion model based on previously edited images. **(3)** The core of our network is the persona-aware 3DGS editing which, by leveraging the personalized diffusion model, respects two predicted attention features-one for editing and the other for ensuring coherence across different viewpoints-allowing for multi-view consistent 3D virtual try-on.

number of tiles it overlaps, with a key assigned to each Gaussian to record view space depth and tile ID. These Gaussians are then sorted by depth, enabling the rasterizer to accurately manage occlusions and overlapping geometry. Finally, a point-based $\alpha$-blend rendering technique is used to compute the RGB color $\mathbf{C}$, by sampling points along the ray at intervals $\delta_i$:

$$\mathbf{C}_{\text{color}} = \sum_{i \in N} \mathbf{c}_i \sigma_i \prod_{j=1}^{i-1} (1 - \sigma_j), \quad \sigma_i = \alpha_i e^{-\frac{1}{2}(\mathbf{x})^{\text{T}} \mathbf{\Sigma}^{-1}(\mathbf{x})}, \tag{2}$$

where $\mathbf{c}_i$ is the color of each point along the ray.

**Instruct-Gaussian2Gaussian (IG2G)** (Cyrus & Ayyan, 2023). Building on Instruct-Pix2Pix (Brooks et al., 2023) and 3DGS, IG2G facilitates text-guided scene editing with a given 3DGS model and its associated training dataset. This process is achieved in two main steps:

*1) Image editing.* For a rendered image from a specified camera viewpoint, IG2G first introduces Gaussian noise to the image. This noisy image, alongside the text embedding $y$ and the original training image, serves as conditions for Instruct-Pix2Pix to generate an edited image, which reflects the desired modifications. These changes will then be back-propagated to the 3DGS scene to update it accordingly.

*2) Dataset update.* In addition to incorporating the editing direction through back-propagation, IG2G updates the entire dataset periodically, specifically every 2,500 training iterations. This update process involves inputting the rendered image into the diffusion model, such as Instruct-Pix2Pix, to ensure stronger and more accurate 3D edits over time.

**Latent Diffusion Model.** Latent Diffusion Model (LDM) (Blattmann et al., 2023b) is a refined variant of diffusion models, optimizing the trade-off between image quality and training efficiency. Specifically, LDM achieves this by first using a pre-trained variational auto-encoder (VAE) (Kingma & Welling, 2013) to project images into a latent space, and then carry out the diffusion process in the latent space. Additionally, LDM enhances the UNet architecture (Ronneberger et al., 2015) by incorporating self-attention mechanisms (Vaswani et al., 2017), cross-attention layers (Vaswani et al., 2017), and residual blocks (He et al., 2016), allowing the model to integrate text prompts as

conditional inputs during the image generation process. The attention mechanism in LDM's UNet is defined as follows:

$$\texttt{ATT}(Q, K, V) = \text{softmax}(\frac{Q \cdot K^T}{\sqrt{d_k}}) \cdot V \tag{3}$$

where $K, Q, V$ represents the key, value, and query features respectively.

## 3.2 PERSONALIZED INPAINTING DIFFUSION MODEL ADAPTATION

Existing methods for editing 3D scenes (Haque et al., 2023; Cyrus & Ayyan, 2023; Wu et al., 2024; Dong & Wang, 2024; Zhuang et al., 2024) typically rely on a pre-trained diffusion model to control the editing process and update the training dataset. However, these approaches would struggle with tasks such as modifying the garment of a human subject (see Fig. 5). A notable cause is that diffusion models like instruct-pix2pix (Brooks et al., 2023) lack the capability to accurately perceive and edit clothing locally. Although there have been advancements in diffusion models (Choi et al., 2024; Zeng et al., 2024; Zhu et al., 2023) for 2D virtual try-on, applying them directly to 3D scene editing often leads to inconsistencies and geometric distortions. This is primarily due to the inherent randomness of diffusion models, which struggle to accurately predict how garments will appear from different viewpoints, leading to discrepancies across various views (see Fig. 3). To tackle this problem in 3D virtual try-on, we propose injecting spatial consistent features derived from the training dataset $\mathcal{I}_{\text{train}}$ into the diffusion model.

**Personalized Diffusion Model via LoRA fine-tuning.** Low-Rank Adaption (LoRA) (Hu et al., 2021) is a technique designed to efficiently fine-tune large language models, and has recently been extended to diffusion models. Rather than adjusting the entire model, LoRA focuses on modifying a low-rank residual component $\Delta\theta$, which is represented as a sum of low-rank matrices. This method allows us to incorporate characteristics of a specific image into the learned distribution of a pre-trained diffusion model.

In order to design an image-prompted network, we first apply LoRA to enhance a pre-trained Stable Diffusion Inpainting Model (Rombach et al., 2022a). Specifically, it involves training the LoRA component $\Delta\theta$ using a collection of edited training images $X_{\text{train}} = \{I_i | i \in [0, n)\}$, where $n$ represents the total number of images, with the following objective:

$$\mathcal{L}(\Delta\theta) = \mathbb{E}_{\epsilon,t}[||\epsilon - \epsilon_{\theta+\Delta\theta}(\sqrt{a_t}\mathbf{z}_{0-i} + \sqrt{1 - a_t}\epsilon, t, y)||^2], \tag{4}$$

where $\mathbf{z}_0 = \mathcal{E}(I_i)$ is the latent embedding from the VAE encoder for image $I_i$, $\epsilon$ is the randomly sampled Gaussian noise, $y$ denotes the text embedding, and $\epsilon_{\theta+\Delta\theta}$ represents the UNet model enhanced with LoRA.

To further enhance the performance, we generate $K$ random binary masks $\mathcal{M} = \{m_i = 0, 1 | i \in [0, K)\}$ and apply these masks to the images (Tang et al., 2024) during LoRA fine-tuning. Then the objective becomes:

$$\mathcal{L}(\Delta\theta_i) = \mathbb{E}_{\epsilon,t}[||\epsilon - \epsilon_{\theta+\Delta\theta}(\sqrt{a_t}\mathbf{z}_{0-i} \odot (1 - m_i) + \sqrt{1 - a_t}\epsilon, t, y)||^2], \tag{5}$$

where $\odot$ denotes the element-wise product.

**Reference-driven Image Editing.** To achieve a well-trained LoRA model, the first critical step is constructing the edited training image set $X_{\text{train}}$. To this end, we further propose reference-driven image editing. Naïvely, one might consider such a straightforward method: applying images from the input human images $\mathcal{I}_{\text{train}}$ directly to a pre-trained 2D virtual try-on diffusion model to obtain the edited images individually. However, we found that this method introduces significant inconsistencies in garment appearance, which adversely affects the quality and

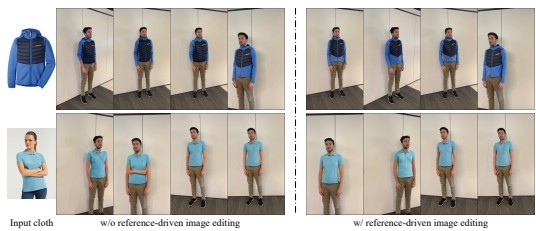

Figure 3: **Effectiveness of reference-driven image editing in multi-view image editing.**

reliability of the LoRA model, as shown in Fig. 3. We attribute this problem to the randomness of the Gaussian noise, which would lead to variations in the attention features.

Drawing inspiration from recent advancements in temporal-aware self-attention techniques used in video generation (Zhou et al., 2024a; Chen et al., 2023a; 2024b; Blattmann et al., 2023a), we propose a novel approach to enhance image consistency using a pre-trained IDM-VTON (Choi et al., 2024). Our approach involves first creating an image set $X_{\text{train}}$ through random sampling of $n$ images from the input multi-view human images $\mathcal{I}_{\text{train}}$. Note that we set $n = 4$ for the experiments reported in this paper. We then perform simultaneous editing of these images while incorporating reference attention features into the denoising process to enhance the overall consistency. Specifically, during the denoising step $t$, we begin by processing the latent features $\mathbf{z}_{t-i}$ of the images $I_i \in X_{\text{train}}$ through the UNet of IDM-VTON, which produces the key and value matrices $K_{t-i}$ and $V_{t-i}$ for the self-attention mechanism. We then integrate reference attention features to update these matrices accordingly:

$$K_{t-i} := [K_{t-i}, K_{t-\text{ref}}], \quad V_{t-i} := [V_{t-i}, V_{t-\text{ref}}], \quad i = 0, ..., n \tag{6}$$

where $[\cdot]$ represents the concatenation operation. In our implementation, we treat the first image as the reference image, i.e., $K_{t-\text{ref}} = K_{t-0}$, $V_{t-\text{ref}} = V_{t-0}$. We then replace the corresponding matrices in the UNet with these updated values to obtain the edited images:

$$X_{\text{train}} := \{F_\theta(I_i, I_{\text{ref}}) | i = 0, ..., n - 1\}, \tag{7}$$

where $F_\theta(\cdot)$ denotes the pre-trained IDM-VTON model. This approach ensures that during the denoising steps, the intermediate latents are influenced by consistent reference features, thereby improving the overall consistency of the edited images.

### 3.3 PERSONA-AWARE 3DGS EDITING

After developing a fine-tuned personalized inpainting diffusion model, integrating it into the 3DGS editing pipeline introduces additional challenges. Unfortunately, images generated by this fine-tuned diffusion model can still exhibit inconsistencies, particularly when the rendered viewpoints differ significantly from those in the edited image set $X_{\text{train}}$. Consequently, this can negatively impact 3DGS editing by introducing visual artifacts and inconsistent textures (see Fig. 7). The problem stems from the limited number of training images used during fine-tuning, which restricts the model's ability to produce consistent features across various viewpoints. This issue remains even when we increase the number of images for LoRA fine-tuning (see Appx. **??**), which also raises GPU memory requirements and reduces training efficiency.

To address this, we propose persona-aware 3DGS editing, which refines diffusion process by merging two predicted attention features: one based on the editing direction and the other derived from the edited image set $X_{\text{train}}$:

$$\text{ATT}(Q_j, K_j, V_j) := \lambda \cdot \text{ATT}(Q_j, K_j, V_j) + (1 - \lambda) \cdot \frac{1}{n} \sum_{i \in X_{\text{train}}} \text{ATT}(Q_j, K_i, V_i), \tag{8}$$

where $\lambda$ is a hyper-parameter to balance the effects, and defaults to 0.55 in our experiments. Instead of adapting the original stable diffusion inpainting model with LoRA, we adapt it via a ControlNet-based stable diffusion inpainting model to condition the inpainting process on the input garment image, thus enhancing the fidelity of the results. Formally, given a rendered image $I_{\text{src}}$ from 3DGS scene and a garment image $I_{\text{cloth}}$ with captioning text $y$ from BLIP-2 (Li et al., 2023), we first input these into the fine-tuned personalized inpainting diffusion model equipped with ControlNet $\mathcal{C}$ to obtain the edited image:

$$I_{\text{edit}} = \epsilon_{\theta + \Delta\theta}(\mathbf{z}_{\text{src}}; y, t, \mathcal{C}(I_{\text{cloth}})), \tag{9}$$

where $\mathbf{z}_{\text{src}}$ represents the encoded latents from the rendered image. Our optimization objective is then be formulated as:

$$\mathcal{L} = \lambda_1 \cdot \mathcal{L}_{\text{MAE}}(I_{\text{edit}}, I_{\text{src}}) + \lambda_2 \cdot \mathcal{L}_{\text{LPIPS}}(I_{\text{edit}}, I_{\text{src}}), \tag{10}$$

where $\lambda_1$ and $\lambda_2$ are hyper-parameters, which defaults to 10 and 15 respectively.

### 3.4 IMPLEMENTATION DETAILS

GS-VTON builds upon official implementation of GaussianEditor (Chen et al., 2023b) for 3DGS editing. While GaussianEditor uses a large language model (Kirillov et al., 2023) to create a 2D image mask and then invert it for labeling locally edited 3D Gaussians, we take a different approach

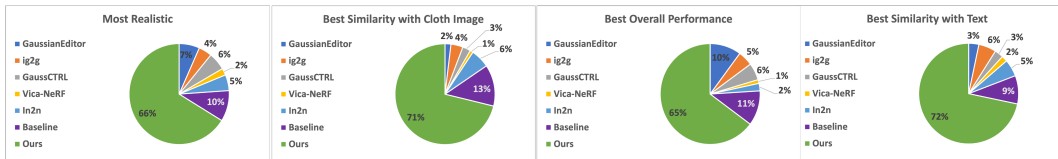

Figure 4: **User study.** Numbers are averaged over 625 responses from 25 volunteers.

by employing a 2D human parsing model (Li et al., 2020) and a human pose estimation model (Güler et al., 2018) to generate the image mask. For our personalized inpainting diffusion model, we utilize the Stable-Diffusion-2-Inpainting model (Stability.AI, 2023b) and adopt hyperparameters from RealFill (Tang et al., 2024). We utilize the pre-trained BLIP-2 model to generate captions for the garment image, which serves as part of the input to the diffusion model. Unlike many existing 3D editing methods that are limited to a maximum image resolution of $512 \times 512$ due to constraints from Instruct-pix2pix, GS-VTON can operate without such limitations, allowing edits at the original resolution of the 3D scene. Additionally, while other methods may adjust hyperparameters for different scenes, we keep all hyperparameters fixed across our experiments. For experiments reported in this paper, we fine-tune the LoRA module for 1,000 iterations, while the 3DGS editing stage involves 4,000 iterations. Typically, the fine-tuning of the LoRA module takes about 30 minutes, and the 3DGS editing requires approximately 25 minutes on a single V100 GPU with 32GB of memory.

## 4 EXPERIMENTS

We now evaluate the performance of our GS-VTON both quantitatively and qualitatively, and provide comparisons with other SOTA methods for 3D scene editing.

**3D-VTONBench.** Existing virtual try-on techniques primarily focus on 2D image generation, while the majority of 3D virtual try-on methods (Rong et al., 2024; Feng et al., 2022; Jiang et al., 2020; Corona et al., 2021; Pons-Moll et al., 2017; Grigorev et al., 2023) are centered around dressing the SMPL models (Loper et al., 2015; Pavlakos et al., 2019) with human garments. On the other hand, current 3D scene editing approaches tend to work with general scenes, leaving 3D virtual try-on underexplored. As a result, there is a notable lack of specific evaluation benchmarks for this task. To thoroughly assess the effectiveness of our methods, we introduce 3D-VTONBench, the first benchmark dataset dedicated to evaluating 3D virtual try-on. Our dataset includes 60 data subjects captured in various poses and garments. We believe that 3D-VTONBench will foster further research in this important area.

**Comparison Methods.** We compare the editing results with five techniques: GaussianEditor (Chen et al., 2023b), Instruct-Gaussian2Gaussian (IG2G) (Cyrus & Ayyan, 2023), GaussCTRL (Wu et al., 2024), Instruct-NeRF2NeRF (IN2N) (Haque et al., 2023), and Vica-NeRF (Dong & Wang, 2024). Since these methods only accept text prompts as input, we use ChatGPT to generate the text prompts corresponding to the clothing images. We don't compare with GaussianVTON (Chen et al., 2024a) as their code is not publicly available.

### 4.1 QUANTITATIVE EVALUATIONS

**User Studies.** We begin by conducting a series of user studies with 25 pairs of edited results to assess the quality of our method. For each pair, we presented the videos generated by our method alongside those from five comparison methods (Chen et al., 2023b; Cyrus & Ayyan, 2023; Haque et al., 2023; Dong & Wang, 2024; Wu et al., 2024). Participants were asked to watch these videos and select the best result based on (1) realism, (2) similarity to the clothing image, and (3) overall performance. A total of 25 volunteers participated in the user studies, providing 625 responses overall. The results, provided in Fig. 4, show that our method significantly outperformed the others across all three dimensions. Furthermore, the evaluation of similarity to the clothing image highlights the limitations of text descriptions in conveying garment details, emphasizing the necessity for our image-prompted pipeline.

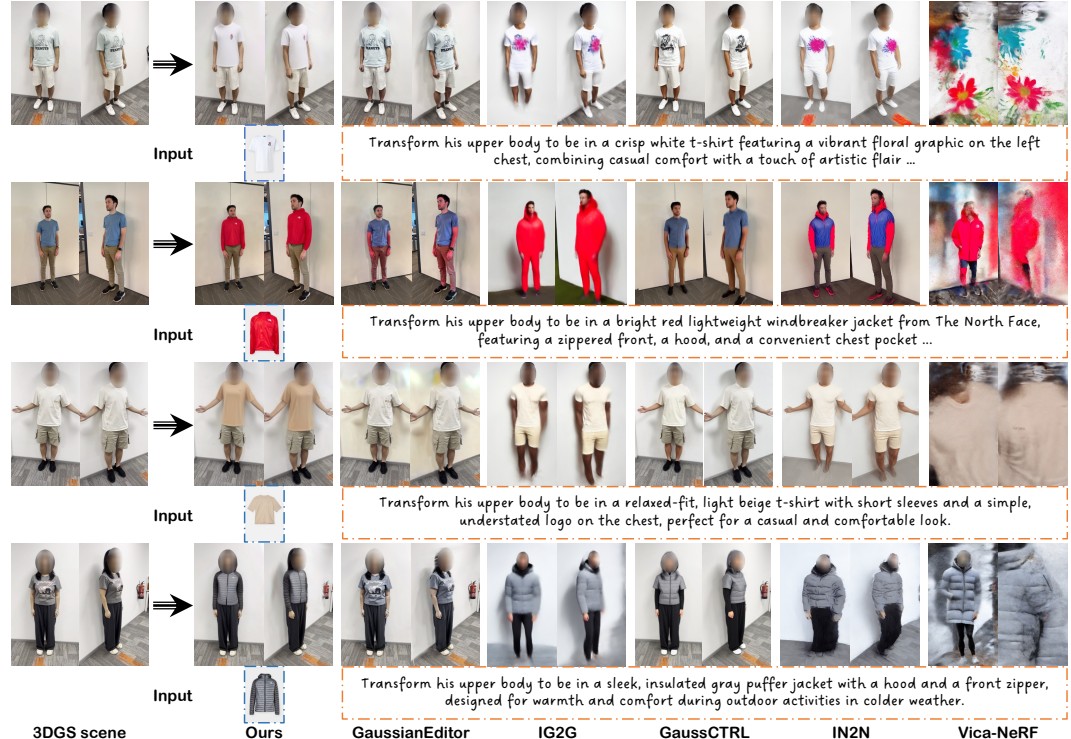

Figure 5: **Qualitative comparison with existing 3D scene editing techniques.** In contrast to other methods that often struggle to produce satisfactory virtual try-on results, our approach consistently delivers high-quality geometry and texture, closely resembling the input garment image.

## 4.2 QUALITATIVE EVALUATIONS

**Comparison with baseline method.** We begin with qualitative evaluations to first compare our approach against the baseline method. Specifically, the baseline method achieves 3D virtual try-on by (1) generating edited training image set $X_{\text{train}}$ individually via IDM-VTON (Choi et al., 2024); (2) fine-tuning LoRA module; (3) editing the 3D scene with fine-tuned model. Results are provided in Fig. 6. The results reveal that the baseline method encounters challenges in three main areas of 3D virtual try-

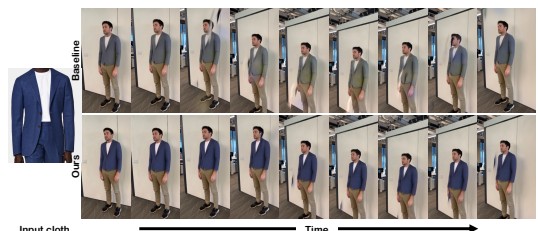

Figure 6: **Comparison with baseline method.**

on: **(1)** it has trouble generating outputs that closely resemble the input garment image; **(2)** it struggles to maintain consistency across different frames; and **(3)** it tends to produce artifacts, such as outliers. In contrast, our contributions, which include reference-driven image editing and persona-aware 3DGS editing, effectively lead to consistent results that align closely with the garment image.

**Comparisons with SOTA methods.** We provide visual comparisons with existing methods in Fig. 5, from which we can draw the following conclusions: **(1)** Textual prompts, even when carefully refined, often struggle to capture the details of garments. This limitation contributes to the tendency of existing methods to produce suboptimal 3D scenes for virtual try-on compared to our approach; **(2)** While GaussianEditor (Chen et al., 2023b) enables local editing using a large language model (Kirillov et al., 2023), it has difficulty making substantial changes to the original geometry and textures. This leads to 3D scenes that do not accurately reflect the textual descriptions; **(3)** GaussCTRL (Wu et al., 2024) utilizes a depth-conditioned ControlNet (Zhang & Agrawala, 2023) to tackle inconsistency issues. However, it struggles with (i) preserving the original identity and

(ii) producing results with insufficient editing; **(4)** Instruct-NeRF2NeRF (Haque et al., 2023) and Instruct-Gaussian2Gaussian (Cyrus & Ayyan, 2023) effectively extract information from text inputs, yet they struggle to (i) keep the background unchanged, (ii) maintain the original identity and poses, and (iii) produce high-resolution renderings; **(5)** Although Vica-NeRF (Dong & Wang, 2024) performs well with general scenes, it has difficulty editing human-centric 3D environments. In contrast, our method consistently produces superior results, offering higher-quality details in both geometry and texture, along with strong consistency with the provided garment image. Additional comparisons can be found in the Appendix.

### 4.3 ABLATION STUDY

**Effectiveness of Persona-aware 3DGS Editing.** We then conduct ablation studies to assess our persona-aware 3DGS editing and the use of ControlNet, with results shown in Fig. 7. Both components are essential for ensuring consistent 3D scene editing; without them, the edited scenes struggle to (1) maintain consistent texture across frames and (2) match the texture of the input garment.

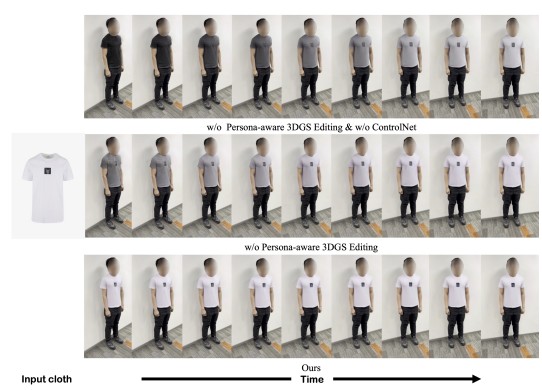

Figure 7: **Analysis of persona-aware 3DGS editing and the utilization of ControlNet.**

**Effectiveness of Reference-driven Image Editing.** In Fig. 8, we present ablation studies to assess the effect of our proposed reference-driven image editing. Existing diffusion models for 2D virtual try-on often demonstrate inconsistencies when editing multi-view images individually (as shown in Fig.3). This inconsistency can hinder the effective fine-tuning of the LoRA module, resulting in subpar 3DGS editing. For instance, the results shown in Fig. 8, edited without our design, display a mismatch in texture with the input garment image. In contrast, our reference-driven image editing effectively addresses this issue, yielding high-fidelity 3D edits with textures that remain consistent with the input.

## 5 CONCLUSION

In this paper, we have introduced GS-VTON, a novel image-prompted method for 3D virtual try-on. We first propose a personalized diffusion adaptation through LoRA fine-tuning, allowing the model to better represent the input garment by extending its pre-trained distribution. Additionally, we introduce reference-driven image editing to enable consistent multi-view editing, providing a solid foundation for LoRA fine-tuning. To further enhance multi-view consistency in the edited 3D scenes, we present persona-aware 3DGS editing, which respects both the desired editing direction and features derived from the original edited images. Extensive evaluations demonstrate the effectiveness of our design, highlighting that GS-VTON delivers high-fidelity results across a range of scenarios and significantly outperforms state-of-the-art methods.

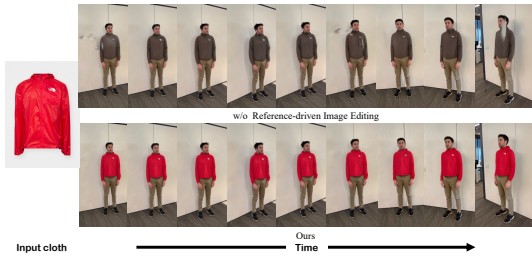

Figure 8: **Effectiveness of reference-driven image editing for 3D virtual try-on.**

**Limitations.** While establishing a new state-of-the-art for 3D virtual try-on, our GS-VTON approach still has some limitations: (1) Inheriting biases from pre-trained 2D virtual try-on models, our pipeline has difficulty accurately modeling long hair when it intersects with clothing. (2) Although our method can accommodate human subjects in various poses, it encounters challenges with severe self-occlusion, such as when a person crosses their arms in front of the chest.

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
