# OpenReview forum: "GS-VTON: Controllable 3D Virtual Try-on with Gaussian Splatting"
_ICLR.cc/2025/Conference — Submitted to ICLR 2025_

### Official Review · Reviewer_W66J · 2024-10-30

**Soundness:** 3
**Presentation:** 3
**Contribution:** 2
**Rating:** 5
**Confidence:** 4

**Summary:**

This paper introduces an image-prompted 3D VTON method (dubbed GS-VTON) which, by leveraging 3D Gaussian Splatting (3DGS) as the 3D representation, enables the transfer of pre-trained knowledge from 2D VTON models to 3D while improving cross-view consistency. Specifically, they propose a personalized diffusion model that utilizes low-rank adaptation (LoRA) fine-tuning to incorporate personalized information into pre-trained 2D VTON models. Moreover, they introduce a reference-driven image editing approach that enables the simultaneous editing of multi-view images while ensuring consistency. Furthermore, they propose a persona-aware 3DGS editing framework to facilitate effective editing while maintaining consistent cross-view appearance and high-quality 3D geometry. Additionally, they proposed a new 3D VTON benchmark, 3D-VTONBench, which facilitates comprehensive qualitative and quantitative 3D VTON evaluations. The experiments demonstrate the superior fidelity and advanced editing capabilities of GS-VTON.

**Strengths:**

1. The paper is well-written and easy to understand.
2. This paper introduces a new perspective on persona-aware editing which effectively improves the performance of the 3D VTON task.
3. The limitations of the proposed method are well-discussed.

**Weaknesses:**

1. There are concerns about the contribution of the proposed module, as it mainly adopts LoRA and a pre-trained diffusion model. For example, the X_train is not explained clearly in Line 339. Does it only refer to the results of the pre-trained IDM-VTON model?
2. I noticed that the paper does not provide evaluation metrics (e.g., LPIPS, FID). Including these metrics would improve the evaluation of the GS-VTON method.
3. There are some writing errors: a) Line 31: "GS-VTONhas" is missing a space. b) Missing commas in Functions 7 and 10.

**Questions:**

1. It is better to discuss the reproducibility.

---

> ### Author Response · Authors · 2024-11-28
>
> # Response to reviewer `#W66J`
>
> > For example, the X_train is not explained clearly in Line 339. Does it only refer to the results of the pre-trained IDM-VTON model?
>
> As mentioned in Line 298, $X_{train}$ is used to fine-tune the personalized diffusion model with LoRA. It's not the results of the pre-trained IDM-VTON model. More specifically, it is derived by expanding the pre-trained IDM-VTON model to enable simultaneous editing of multiple images, integrating the attention features outlined in Eq. (6).
>
>
> > I noticed that the paper does not provide evaluation metrics (e.g., LPIPS, FID). Including these metrics would improve the evaluation of the GS-VTON method.
>
> We wanted to conduct such comparisons. However, these metrics require ground truth for calculation, which is unavailable in the 3D VTON setting we explore.
>
> Additionally, we conducted a CLIP-based direction score (CLIP-DS) and a CLIP-based score (CLIP-S) to compare with existing techniques. The results are provided below, which demonstrate the superiority of our method.
>
> |            | GaussianEditor |   IG2G   | GaussCTRL |   IN2N   | Vica-NeRF | Baseline  |   Ours    |
> |:----------:|:--------------:|:--------:|:---------:|:--------:|:---------:|:---------:|:---------:|
> | **CLIP-DS**|      7.90       |  15.33   |   19.01   |  13.38   |   8.99    |   17.91   | **22.18** |
> | **CLIP-S** |     21.71       |  17.92   |   19.33   |  17.04   |  16.76    |   22.19   | **27.19** |
>
>
>
> > There are some writing errors: a) Line 31: "GS-VTONhas" is missing a space. b) Missing commas in Functions 7 and 10.
>
> We have rectified the writing errors in the paper. Thank you for the careful review!
>
> > It is better to discuss the reproducibility.
>
> We have strived to ensure that our implementation details are thorough when introducing the method and writing the implementation details. Meanwhile, our code will be released shortly for reproduction of our results.

---

### Official Review · Reviewer_bmEM · 2024-10-31

**Soundness:** 3
**Presentation:** 3
**Contribution:** 3
**Rating:** 6
**Confidence:** 5

**Summary:**

The paper introduces a novel image-prompted 3D virtual try-on method that leverages 3D Gaussian Splatting for fine-grained editing of human garments within a 3D scene. The authors propose a personalized diffusion model adapted via LoRA fine-tuning to incorporate personalized information into pre-trained 2D VTON models. They also introduce a persona-aware 3DGS editing framework to maintain consistent cross-view appearance and high-quality 3D geometry. The paper establishes a new benchmark, 3D-VTONBench, for comprehensive 3D VTON evaluations and demonstrates through extensive experiments that the proposed GS-VTON method outperforms existing techniques in terms of fidelity and editing capabilities.

**Strengths:**

* The paper presents a groundbreaking method for 3D virtual try-on by extending pre-trained 2D VTON models to 3D using 3DGS, addressing the challenge of cross-view consistency and spatial relationships in 3D scenes.
* The establishment of the 3D-VTONBench dataset is a valuable resource for the research community, facilitating more comprehensive evaluations and fostering further advancements in 3D VTON.
* The method demonstrates superior performance over existing techniques.

**Weaknesses:**

* In some cases, such as the first row in Fig. 1, there are noticeable artifacts on the sleeves and edges of the garments.

* The statements about the effects of persona-aware 3DGS editing are inconsistent between the abstract and introduction. The abstract states "maintain consistent cross-view appearance," while the introduction says "enhancing multi-view consistency."
* What are the differences or advantages of your methods compared to RelFill in "Personalized Diffusion Model via LoRA fine-tuning"? Some experiments may be needed to demonstrate this.
* The hyperparameter in persona-aware 3DGS editing seems tricky.
* To demonstrate the ability to maintain 3D consistency, the following works should be discussed.

  * Geometry-Aware Score Distillation via 3D Consistent Noising and Gradient Consistency Modeling
  * MaTe3D: Mask-guided Text-based 3D-aware Portrait Editing
  * ConsistNet: Enforcing 3D Consistency for Multi-view Images Diffusion
* Need more detailed descriptions of the proposed benchmark.

**Questions:**

* In line 110, does LoRA "extend its learned distribution"? Are there any citations?
* I need more details about ControlNet. Is it from the original repository?

---

> ### Author Response · Authors · 2024-11-28
>
> # Response to reviewer `#bmEM`
>
> > In some cases, such as the first row in Fig. 1, there are noticeable artifacts on the sleeves and edges of the garments.
>
> Thanks for pointing this out. From our perspective, the results are satisfactory and notably superior to current techniques. Moreover, we have noted that increasing the batch size during training can be effective in mitigating this particular issue.
>
>
> > The statements about the effects of persona-aware 3DGS editing are inconsistent between the abstract and introduction. The abstract states "maintain consistent cross-view appearance," while the introduction says "enhancing multi-view consistency."
>
> Thanks for pointing out. To maintain objectivity and avoid being absolute, we have modified the statement to "enhance multi-view consistency" in the paper.
>
> > What are the differences or advantages of your methods compared to RelFill in "Personalized Diffusion Model via LoRA fine-tuning"? Some experiments may be needed to demonstrate this.
>
> We have provided related comparisons in Fig. 3. Specifically, "w/o reference-driven image editing" refers to RealFill, which shows inconsistencies across different views.
>
> > The hyperparameter in persona-aware 3DGS editing seems tricky.
>
> We have conducted experiments with various value of the hyperparameter λ (used in Eq. (8)) and observed that our method's performance is robust. Additionally, for the experiments presented in the paper, we used a fixed λ value to eliminate variability caused by differing hyperparameters.
>
> > To demonstrate the ability to maintain 3D consistency, the following works should be discussed.
>
> The GSD (Geometry-aware Score Distillation) technique introduces a 3D-consistent noise strategy to enhance text-to-3D generation, addressing a distinct task compared to our method. ConsistNet also prioritizes multi-view consistency to enhance the image-to-3D pipeline. MaTe3D, on the other hand, is a mask-guided and text-based framework for 3D-aware portrait editing utilizing GANs and diffusion models, albeit necessitating a comprehensive dataset for training.
>
> We will cite and include discussions with these papers in the revised edition.
>
> > In line 110, does LoRA "extend its learned distribution"? Are there any citations?
>
> Yes. We would like to direct the reviewer's attention to the LoRA paper [1].
>
> > I need more details about ControlNet. Is it from the original repository?
>
> Yes. The ControlNet implementation utilized in our research is based on the original repository, with the weights being specifically sourced from Edit-Anything-v0-3.
>
> [1] LoRA: Low-Rank Adaptation of Large Language Models. ICLR 2022

---

### Official Review · Reviewer_2fdr · 2024-11-02

**Soundness:** 3
**Presentation:** 1
**Contribution:** 3
**Rating:** 5
**Confidence:** 4

**Summary:**

This study proposes GS-VTON, a novel and potentially first-of-its-kind 3D virtual try-on method, based on a conditional diffusion model and 3D Gaussian Splatting. Unlike traditional 2D virtual try-on methods, GS-VTON enables users to visualize how clothing would appear on their bodies from different viewing angles, making it particularly promising for VR/AR applications. Moreover, the proposed reference-driven image editing and persona-aware 3D Gaussian Splatting techniques improves multi-view consistency in the virtual try-on experience in 3D.

**Strengths:**

1. GS-VTON is the first 3D virtual try-on method, showing more diverse real-world applications compared to 2D virtual try-on. It holds promising potential to transform online shopping and create positive social impact.
2. GS-VTON utilizes Reference-driven Image Editing and 3D Gaussian editing to ensure the try-on scene is consistent in both texture and geometry across multiple views. The design seems sound.

**Weaknesses:**

1) **Reference-driven Image Editing**: The authors propose this method to ensure texture consistency across multi-view images by integrating attention features from a reference image. However, if the reference image has incorrect textures, it may negatively affect the consistency of subsequent images.

2) **Questionable Experimental Setting**:
   - All benchmark methods use text as input, while GS-VTON uses an image as a prompt. However, the user study criterion of clothing image similarity may not be ideal for comparing these approaches.
   - Though the authors provide qualitative examples comparing GS-VTON with a baseline 2D VTON method, including this baseline in the user study for comprehensive quantitative analysis is essential. Most of the compared methods were not natively designed for virtual try-on applications, raising concerns about experimental fairness.

3) **Limited View Coverage**: GS-VTON mainly shows the front of the clothing, without displaying how the back of the body would appear.

4) **Pipeline Presentation Issue**: Figure 2 shows Reference-driven Image Editing as the first step, followed by the Personalized Diffusion Model via LoRA Fine-tuning. However, the main text introduces these components in reverse order, which caused some confusion initially.

**Questions:**

1. Could the authors provide more details on how the $G_{src}$ point cloud was collected in Figure 2? Since RGB-D sensors or other 3D sensors are less accessible compared to standard cameras, this may limit the method's real-world applicability. A more practical scenario might involve capturing multi-view images with a camera, but in that case, the lack of camera pose information could limit the feasibility of training the 3D GS model.

2. Could the authors provide more information about the response processing in the user study? Was a crowdsourcing platform used for data collection?

---

> ### Author Response · Authors · 2024-11-28
>
> # Response to reviewer `#2fdr`
>
>
> > However, if the reference image has incorrect textures, it may negatively affect the consistency of subsequent images.
>
> In Eq. (6) (Lines 226-338), the reference image is defined as the first input image, implying that it should inherently be devoid of any extraneous textures or anomalies. Additionally, our experimental results demonstrate that the reference-driven image editing process consistently generates accurate and coherent textures. These clarifications will be incorporated into the revised PDF, and we are preparing to release the code shortly.
>
> > All benchmark methods use text as input, while GS-VTON uses an image as a prompt. However, the user study criterion of clothing image similarity may not be ideal for comparing these approaches.
>
> > Though the authors provide qualitative examples comparing GS-VTON with a baseline 2D VTON method, including this baseline in the user study for comprehensive quantitative analysis is essential.
>
> We are grateful for the suggestion, and as a result, we have extended our user studies to include the baseline method and consider text-similarity aspects during the rebuttal phase. The results of these assessments are outlined in Fig.4, demonstrating the superiority of our approach.
>
>
> > GS-VTON mainly shows the front of the clothing, without displaying how the back of the body would appear.
>
> We mainly showcase the front of the clothing for various reasons: (1) Cloth details are typically focused on frontal perspectives in many instances; (2) The input cloth images provided are only frontal views; (3) The human dataset obtained from Instruct-NeRF2NeRF exclusively comprises frontal views, which we follow when structuring our datasets.
>
> > Figure 2 shows Reference-driven Image Editing as the first step, followed by the Personalized Diffusion Model via LoRA Fine-tuning. However, the main text introduces these components in reverse order, which caused some confusion initially.
>
> We appreciate the suggestion provided by the reviewer, and as a response, we have updated the pipeline figure within the main paper.
>
> > Could the authors provide more details on how the G_src point cloud was collected in Figure 2?
>
> We obtain the $G_{src}$ point cloud directly from the pre-trained 3D Gaussian splitting model using the code provided by 3DGS[1]. The visualization depicted in Fig. 2 is captured using MeshLab.
>
> Our approach does not rely on calibrated cameras typically used with RGB-D or 3D sensors as inputs. Instead, it operates seamlessly with uncalibrated multi-view images. Camera calibrations are acquired using COLMAP[2]. In contrast to techniques utilizing RGB-D data, our method offers a more versatile solution suitable for various everyday applications.
>
> > Could the authors provide more information about the response processing in the user study? Was a crowdsourcing platform used for data collection?
>
> A screenshot of our user studies is provided in Fig. 11. These studies were carried out through a questionnaire format on a crowdsourcing platform. A total of 25 volunteers took part in the study, representing a diverse group consisting of animators, AI researchers, and gaming enthusiasts, with ages ranging from 20 to 35.
>
>
> [1] 3D Gaussian Splatting for Real-Time Radiance Field Rendering. Siggraph 2023
>
> [2] Structure-from-Motion Revisited. CVPR 2016

---

> ### Comment · Reviewer_2fdr · 2024-12-02
>
> I appreciate the authors' detailed response. However, I still have a bit concerns regarding the feasibility of GS-VTON in real-world settings. The method requires COLMAP to estimate camera parameters from uncalibrated multi-view images, which can be time-consuming. The method need to re-initialize the point cloud and update the Gaussian parameters for each set of multi-view images, resulting in high latency during inference. I suggest the author to try more efficient point cloud initialization method like Dust3R[1] in the future work. I therefore maintain my rating at current status.
>
> [1] Wang, Shuzhe, et al. "Dust3r: Geometric 3d vision made easy." Proceedings of the IEEE/CVF Conference on Computer Vision and Pattern Recognition. 2024.

---

> ### Author Response · Authors · 2024-12-02
>
> Thank you again for your constructive feedback.
>
> We will consider applying more efficient point cloud initialization as in Dust3R in the future works. However, we argue that:
>
> **(1)** The utilization of COLMAP is not one of our contributions or the focus of this work. Meanwhile, we can apply other methods to obtain the camera calibrations. In this work, we choose to follow all the existing 3D scene editing techniques (Instruct-NeRF2NeRF[1], GaussianEditor[2], GaussCtrl[3], Vica-NeRF[4]) and even the original 3D Gaussian Splatting[5] to employ COLMAP for extracting camera calibrations from uncalibrated multi-view images.
>
> **(2)** It's worth mentioning that COLMAP only requires application once for each set of data. Subsequently, we can perform various editing directly, making it more efficient for 3D scene editing frameworks.
>
> [1] Instruct-NeRF2NeRF: Editing 3D Scenes with Instructions. ICCV 2023
>
> [2] GaussianEditor: Swift and Controllable 3D Editing with Gaussian Splatting. CVPR 2024
>
> [3] GaussCtrl: Multi-View Consistent Text-Driven 3D Gaussian Splatting Editing. ECCV 2024
>
> [4] ViCA-NeRF: View-Consistency-Aware 3D Editing of Neural Radiance Fields. NeurIPS 2023
>
> [5] 3D Gaussian Splatting for Real-Time Radiance Field Rendering. Siggraph 2023

---

### Official Review · Reviewer_23CD · 2024-11-12

**Soundness:** 3
**Presentation:** 3
**Contribution:** 3
**Rating:** 6
**Confidence:** 2

**Summary:**

This paper introduces a novel approach for achieving 3D virtual try-on (VTON) that addresses current limitations in consistency and spatial relationships when extending 2D VTON methods to 3D. The method, GS-VTON, leverages 3D Gaussian Splatting (3DGS) as a 3D representation framework, combined with personalized diffusion model adaptation using LoRA (Low-Rank Adaptation) fine-tuning. This allows for multi-view image editing with consistency across different viewpoints and high-quality geometric and texture fidelity. Additionally, the paper introduces a benchmark, 3D-VTONBench, to support quantitative and qualitative evaluations for 3D VTON methods. The experiments show that GS-VTON outperforms state-of-the-art techniques, establishing a new benchmark for 3D VTON performance.

**Strengths:**

+ Effectively bridges the gap between 2D VTON and 3D applications by incorporating 3D Gaussian Splatting, which ensures consistency across multi-view images.
+ Uses a personalized diffusion model with LoRA fine-tuning, improving adaptability and customization for different subjects and garments.
+ Presents a new benchmark, 3D-VTONBench, which is an important addition for the comprehensive evaluation of 3D VTON performance.
They also demonstrates superior performance over existing methods, particularly in areas of realism, garment detail accuracy, and editing consistency.

**Weaknesses:**

- The model is a very straightforward follow-up of 2D VTON models and inherits some biases from pre-trained 2D VTON models.

**Questions:**

From line 62-68, is the pink stuff on the rightmost edited images because of the artifacts of the proposed method?

Personally, I think this paper is a good follow-up on 2D virtual try-on method and will be beneficial to this research area. Extensive experiments and comparisons with state-of-the-art techniques demonstrate the superiority of GS-VTON in terms of realism and multi-view consistency. This is validated not only through quantitative metrics but also through user studies. Therefore,  I don't have any major concerns.

---

> ### Author Response · Authors · 2024-11-28
>
> # Response to reviewer `#23CD`
>
> > The model is a very straightforward follow-up of 2D VTON models and inherits some biases from pre-trained 2D VTON models.
>
> We concur with the reviewer's point that our method could potentially carry biases from the pre-trained 2D VTON models, influenced by the constraints of the training dataset. Nevertheless, we wish to highlight that our personalized inpainting diffusion model adaptation is designed to significantly reduce the adverse effects of these biases.
>
> > From line 62-68, is the pink stuff on the rightmost edited images because of the artifacts of the proposed method?
>
> Thanks for pointing this out. Indeed, the presence of the pink artifacts is a result of certain artifacts. Moreover, we have observed that increasing the batch size during training can help address this issue.

---

> > ### Comment · Reviewer_23CD · 2024-11-30
> >
> > I thank authors for the response and I thus maintain my score.

---

### Author Response · Authors · 2024-12-01

We thank all reviewers for their time and effort in reviewing our paper!

Our method presents the first method for text-guided 3D virtual try-on (VTON) methods via diffusion models. Our method proposes two major components to realize this goal:

**(1)** We observe that images offer richer and more precise information compared to text prompts. To this end, we propose to leverage low-rank adaptation (LoRA) to integrate image priors. Additionally, we introduce a reference-driven image editing technique that enables the simultaneous editing of multi-view images while maintaining consistency, thereby enhancing the training of the LoRA module.

**(2)** We propose a persona-aware 3DGS editing framework for 3D virtual try-on. This framework involves adjusting the attention module to ensure a consistent appearance across multiple views during the editing process.

**Below, we summarize the changes made according to the reviews:**

1. We explore the impact of biases inherited from the pre-trained diffusion model on our method. Furthermore, we discuss how fine-tuning the LoRA module in our framework can help mitigate and improve this scenario (`#23CD`).

2. We discuss the pink artifacts presented in Fig.1 and how we can address this issue (`#23CD`, `#bmEM`).

3. We discuss the definition of the reference image used in Eq. (6) and how our reference-drive image editing will positively improve the consistency of the subsequent images (`#2fdr`).

4. We conduct user studies to further compare with the baseline method and include the metric of text similarity. We also provide more discussion about the user studies. (`#2fdr`).

5. We analyze and discuss our data with only frontal views (`#2fdr`).

6. We update the pipeline figure to make it more consistent with the writing (`#2fdr`).

7. We provide more details about how we obtain the 3D Gaussian points from multi-view input images (`#2fdr`).

8. We update the statement of the claims in the paper to make them more consistent (`#bmEM`).

9. We show comparisons with RealFill (`#bmEM`).

10. We analyze the hyper-parameters used in the persona-aware 3DGS editing (`#bmEM`).

11. We discuss more related works suggested by the reviewer to ensure comprehensive discussion(`#bmEM`).

12. We illustrate more details about the LoRA and ControlNet used in our framework (`#bmEM`).

13. We further clarify how we obtain the X_train via IDM-VTON (`#W66J`).

14. We conduct quantitative evaluations via CLIP to further demonstrate our method (`#W66J`).

15. We improve our writing to rectify the typos and discuss more about the implementation details (`#W66J`).

We sincerely thank all reviewers and the AC(s) again for their valuable suggestions, which have significantly contributed to enhancing the quality of our paper.

If you have any further questions, we would be happy to discuss them!

---

### Meta-Review · Area_Chair_nwbQ · 2024-12-19

**Metareview:**

This submission proposes a 3D virtual try-on method that leverages 3D Gaussian Splatting (3DGS) and diffusion model adaptation to address limitations in consistency and spatial relationships, present in existing 2D methods. The approach enables multi-view image editing towards improving consistency across different viewpoints. The paper also establishes a new benchmark, 3D-VTONBench, for evaluating 3D VTON techniques. Experimental work evidences that the introduced approach can outperform alternative methods in terms of performance and editing capabilities.

Due to the extent of wide-spread proposed changes, resulting from the rebuttal, the manuscript likely benefits from deeper edits (and polish; typographical errors still exist). After reviewing the paper, rebuttal and resulting discussion AC believes that this submission can be strengthened by further refinement and subsequent round of reviews and here recommends rejection. The data contribution is likely of value and may be well suited for a dedicated dataset benchmark track.

**Additional Comments On Reviewer Discussion:**

The paper received four reviews resulting in: two borderline accepts and two borderline rejects.

Reviewers comment on positive aspects related to the nature of the methodology, introduction of a benchmark, sound design and limitations discussion. Negative review comments raised important concerns pertaining to the limited experimental setup, the straightforward follow-up nature of the work (with respect to analogous 2D models), contribution misunderstandings, missing experimental comparisons, absent related works and incomplete benchmark details. Smaller queries related to hyperparameter tuning, lack of quantitative metrics, reference-driven image editing, language inconsistency, writing errors, presentation issues and result quality.

The submission can be considered somewhat borderline, lacking decisive scores. The rebuttal attempts to address concerns however it does not persuade negative reviewers; they remain unconvinced and opt to retain a negative view on the paper citing in particular contribution and feasibility related concerns. Multiple author statements in the rebuttal would have benefited from being made tighter and evidence based. (e.g. "the presence of the pink artifacts is a result of certain artifacts. Moreover, we have observed that increasing the batch size during training can help address this issue"). Authors summarise a large set of manuscript proposed changes, in response to reviewer comments, some of which can be considered non-trivial (e.g. impact of inherited biases, frontal views).

---

### Decision · Program_Chairs · 2025-01-22

Reject